# City-Region Food Systems and Biodiversity Conservation: The Case Study of the Entre-Douro-e-Minho Agrarian Region

Mariana Filipe [1,*], Angela Lomba [1,2], João Pradinho Honrado [1,2] and Andreia Saavedra Cardoso [1]

1   CIBIO, Centro de Investigação em Biodiversidade e Recursos Genéticos, InBIO Laboratório Associado, BIOPOLIS Program in Genomics, Biodiversity and Land Planning, Campus de Vairão, 4485-661 Vairão, Portugal
2   Departamento de Biologia, Faculdade de Ciências, Universidade do Porto, 4099-002 Porto, Portugal
*   Correspondence: marianafilipe@cibio.up.pt

**Abstract:** Agriculture is the dominant form of land management with at least half of the species in Europe depending on agricultural habitats. Additionally, there is a growing demand for a more sustainable food system. In that context, food system relocalization and City-Region Food Systems (CRFS) are proposed for food resilience and environmental sustainability. This work represents the first attempt to map the relocalization of the potential foodshed (PF) of the Entre-Douro-e-Minho agrarian region, assessing its impacts on landscape heterogeneity and ecological value. The methodological approach, developed in a Geographic Information System, aimed to (1) map the ecological suitability of the study area, (2) propose a PF relocalization scenario, and (3) assess its impacts on land cover changes and landscape structure through landscape metrics. Outcomes of this research reflect land-use optimization concerning ecological suitability for agrarian uses, depicting the strong presence of temporary crops in the landscape. They also emphasize the need for greater detail in Land Use Plans, due to the vulnerability of coastal areas. Moreover, results revealed an increased landscape heterogeneity and related ecological value, highlighting the integration of landscape ecological properties into CRFS planning as a line of research and contributing to the implementation of land use compatible with biodiversity conservation.

**Keywords:** food relocalization; foodshed; ecological suitability; landscape heterogeneity; GIS; patch analyst

## 1. Introduction

Agriculture is a major form of land management, representing approximately 40% of the global land area, with food production and consumption patterns impacting landscapes and societies directly [1]. Agricultural landscapes and underlying farming systems are crucial to meet key sustainable development goals, such as food security and environmental sustainability [2,3]. Regardless, over the past 50 years, with population growth and increased agricultural production, the efficiency of the global food system (FS) has declined, with detrimental impacts on the environment, such as greenhouse gases emissions, misuse and pollution of natural resources, degradation of ecosystems, and loss of biodiversity [3]. Additionally, there is a growing perception of the increased vulnerability of the global FS, whether due to climate change, the loss of food production capacity, increased global competition, or the interruption of logistics systems [3,4]. Other events, such as the COVID-19 pandemic or the recent war in Europe, have further exposed the vulnerability of the existing FS [4]. The global FS is undoubtedly out of balance and understandably; there is an urgent call for transformation from researchers, policymakers, and civil society, but redesigning it is a complex endeavor [5].

Currently, there are several perspectives on how to achieve FS sustainability. The transformation perspective advocates FS relocalization, or regionalization, as a structural

change for building new sustainable relationships between cities and the regions they integrate [5,6]. For this reason, City-Region Food Systems (CRFS) and food self-sufficiency (FSS) are increasingly debated, considering the city and its peri-urban and rural perimeter as a functional and spatial continuum, focusing on rural–urban connections in terms of social, functional, and agroecological interactions [7–10]. A sustainable and resilient CRFS aspires to improve regional FSS while providing a vibrant regional food economy and increased resilience by reducing dependence on distant sources of supply [7,11], thus claimed as a strategy to promote land use sustainability and food security [12]. Likewise, CRFS provides a cross-cutting space to foster Sustainable Development Goals (SDGs), playing a central role in achieving local and global sustainability [7]. However, due to the expansion of urban areas, urban and peri-urban agriculture are under threat, and relocalization will depend, at least in part, on agricultural land located close to urban areas [13,14].

The foodshed concept is increasingly used to discuss the geography of urban food supply [10]. Such a concept has emerged in regional planning as an approach to assess the potential ecological productive capacity of a given region or city for food provision regarding its population demand [10]. Foodshed capacity studies provide information on the area necessary to reach levels of theoretical self-sufficiency in a specific geographical unit, halting the loss of agricultural land and allowing for increased and diversified agricultural production [10,15]. Several foodshed analyses have been performed globally and in Europe [10], identifying the potential for regional food self-sufficiency, frequently in main metropolitan areas, and proposing changes in food consumption patterns, land use, and production models [1–3,16,17]. Overall, results demonstrated that FS relocalization is feasible to a certain degree and is dependent on the area's biophysical characteristics, population density and distribution, and food consumption patterns. Hence, this paper addresses the question of whether it is possible to reconcile agricultural production and biodiversity conservation improving food self-sufficiency at the regional scale.

Relocalization through CRFS has also been highlighted as a potential direction towards environmental sustainability [7,11,17,18]. In that sense, CRFS management of ecosystems and natural resources advocates agroecological diversity and may foster biodiversity and the delivery of ecosystem services such as pollination, pest control, and climate resilience as studies on urban agriculture attest [19]. FS relocalization is also pointed out as a way to internalize the negative outcomes of the current food regime as the environmental impacts are becoming even more disconnected from their places of consumption [20]. So far, research on CRFS and foodshed analysis and mapping only considered the environmental impacts from the point of view of land use sustainability [17] and climate change mitigation, e.g., by studying the reduction of carbon footprint through adaptation of diets and decrease in "food miles" [9,13,21] or changing to alternative farming systems, e.g., organic farming [16,22,23] and the closure of biogeochemical cycles [24]. Nevertheless, studies based on Life Cycle Analysis have already criticized relocalization claiming transport makes a relatively minor contribution to overall food chain impacts (about 10–15% of greenhouse gas emissions) [25] and arguing that food miles are a poor indicator of FS sustainability [26]. However, the relocalization of FS as a means to achieve sustainability has been asserted recently with further arguments. This is an innovative research topic for which transdisciplinary methodologies are being developed and applied [27]. Among these arguments is the assumption that land cover patterns diversification (i.e., heterogeneity) possible in local FS positively affects "landscape ecology patterns and processes, and species richness" [27]. Thus, with this research, we intended to move beyond the state-of-the-art by assessing the consequences of relocalization of FS, namely by studying the relationship between land-use changes and their impact on landscape ecological value and linked biodiversity. In this context, the ecological value concept aims to capture the intrinsic value of the landscape, which is specifically linked to the benefits of biotic or abiotic components for the maintenance of organisms, i.e., the benefits that nature provides to sustain life and biodiversity [28–30].

A relevant share of biodiversity in Europe depends on agricultural habitats [31,32], i.e., at least 50% of species [33–35]. Thus, preserving biodiversity in agricultural landscapes

is one of the main goals in reversing the biodiversity decline, as stressed by the European political agenda (i.e., CAP, Green new deal, Farm to Fork strategy, EU Biodiversity Strategy). Also, previous research has demonstrated land use and biodiversity are interdependent, and any change in the former can affect biodiversity reducing related ecosystem services [36]. In fact, widespread land abandonment is endangering the long-term conservation of biodiversity, habitats, and valued landscape by decreasing habitat diversity, simplifying landscape mosaics in their spatial heterogeneity, leading to declines or local extinctions of several flora and fauna species associated with farmland, and increasing the frequency of fires [37–39]. Additionally, in many European landscapes, it has been pointed a positive relationship between the spatio-temporal heterogeneity of ecosystems and local biodiversity [40–43]. Therefore, restoring heterogeneity can be paramount to preserving biodiversity in structurally simple landscapes. For example, at the landscape scale, there is a positive relationship between butterfly abundance and heterogeneity. Similarly, habitat diversity is associated with a greater amount of bird species and an increased variety of generalist insects in plantations [34].

Further, recent research demonstrated that agricultural landscapes under low-intensity agricultural management (i.e., low levels of fertilizers, agrochemicals, and mechanization, with reduced animal density and frequent rotating land uses) can foster a high level of biodiversity and provide a diverse set of ecosystem services [35,44–46]. In the EU, these "traditional" agricultural landscapes are classified as High Nature Value Farmland (HNVf) [47,48]. These areas are well adapted to local climatic, geographic, and environmental conditions, expressing the relationship(s) between farming systems and practices, as well as habitats and species of high conservation value [45,49,50]. Three different types of HNVf have been broadly identified [48]. HNVf with a high presence of semi-natural agricultural habitats was defined as type 1 (hereafter HNVf1). The presence of a mosaic landscape of low-intensity agriculture with natural and structural elements such as field edges, hedges, stone walls, patches of forest, or scrubs is defined as type 2 (hereafter HNVf2). Finally, the presence of species of high conservation interest in often intensively managed landscapes was defined as HNVf type 3 (hereafter HNVf3). Estimates highlight that around 30% of all agricultural land in the EU corresponds to HNVf, which is seen as a critical contribution to meeting biodiversity targets, specifically in protecting species and habitats, while achieving more sustainable agriculture [51].

*Main Goals and Findings*

Foodshed studies have made limited progress regarding the environmental impacts of FS relocalization, and, in particular, there is a knowledge gap in what concerns biodiversity and ecosystem conditions. Further extending previous work, this research aims to address whether improved food self-sufficiency can be achieved on a regional scale while reconciling agricultural production with biodiversity conservation. We proposed a novel method complementing CRFS and foodshed approaches with biodiversity impact analysis, such as the relocalization scenario's impact on the ecological value of the landscape through the assessment of spatio-temporal changes in land use and landscape structure (i.e., heterogeneity). We also integrated the impact assessment on HNVf, which are crucial landscapes for biodiversity conservation.

The agrarian region of Entre-Douro-e-Minho (EDM), in northern Portugal, was selected as our study area, matching the ideal level of geographical disaggregation for a foodshed study. The EDM includes a great diversity of farming systems, integrating the second metropolitan area in the country with significant urbanization pressure and farmland loss. Thus, we believe it is paramount to develop research to understand how agricultural land uses can be planned and designed to minimize biodiversity loss.

The outcomes of this research disclosed a potential scenario reflecting land-use optimization concerning ecological suitability for agrarian uses (agricultural and forestry land uses). However, it highlighted the need for greater detail in subsequent Land Use Plans and other methodologies that investigate the impacts of the agricultural land use

changes in coastal areas, considering their specific ecosystems and vulnerabilities. Further, impacts on the landscape structure indicate an increased landscape heterogeneity and related ecological value, suggesting CRFS planning could be a key strategy to improve landscape planning and land use dynamics towards more sustainable agriculture and biodiversity conservation.

## 2. Materials and Methods

### 2.1. The Study Area

The EDM agrarian region (Figure 1) is located in Northwest Portugal and occupies an area of approximately 877,236.5 ha, encompassing five administrative sub-regions (updated with NUTS III of 2013[1]), namely four inter-municipal communities (Cávado, Ave, Tâmega e Sousa, and Alto Minho) and the Oporto Metropolitan Area (AMP), representing a total of 52 municipalities [52]. The EDM presents a heterogeneous landscape including a mixture of plain areas, large and narrow valleys, and mountain massifs, with altitudes that reach 1500 m. The topography is closely related to the main river basins: Minho, Lima, Cávado, Ave, and Douro [53]. The climate is affected by the proximity of the Atlantic Ocean and the vigorous terrain elevation in the interior. Temperatures are mild, and the thermal amplitudes are less pronounced on the coast than in the interior regions, revealing continental climate characteristics. Additionally, this region has the highest rainfall levels in the country, particularly in the high interior mountains [53].

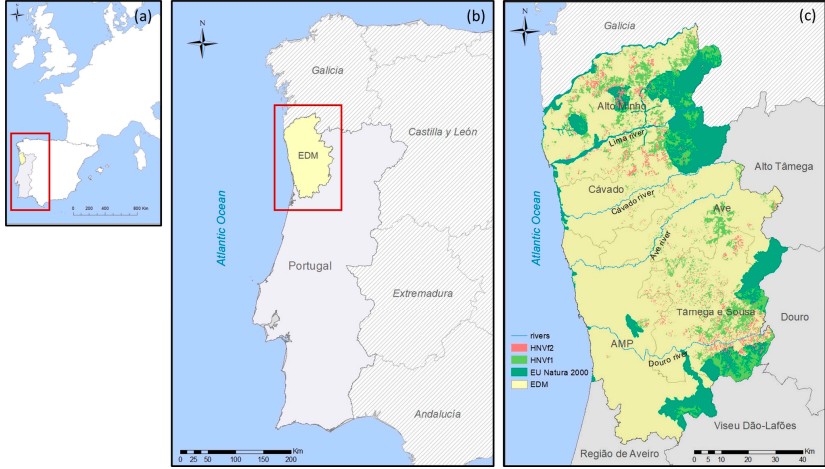

**Figure 1.** The geographic location of the study area, the Entre-Douro-e-Minho (EDM) agrarian region, in (**a**) European, (**b**) Iberian Peninsula, and (**c**) regional contexts. The location of the Natura 2000 areas and High Nature Value farmlands type 1 and 2 (HNVF1, HNVf2, respectively) are also represented.

The EDM is the second-largest urban agglomeration in Portugal concerning population and economic activity. Population density is higher than the national average, with 3,213,026 million inhabitants representing 1/3 of the mainland population [52]. Located in the AMP area of influence, land use allocated to agricultural practices is in constant and unequal competition with other urban land uses [54]. The majority of the territory, 43.6%, corresponds to forest area, with only 22.8% corresponding to utilized agricultural area [55]. Regardless, EDM includes the most relevant dairy production in the country, represented by the concentration of specialized farms in the production of milk and forage cereals [56]. These areas under intensive agricultural management exist in the most fertile plains, in contrast to small-scale and heavily parceled low-intensity farms in the higher lands, which are less fertile [45] and where we can encounter the HNVf. The HNVf1 are dominant in the EDM, and they include the high-altitude irrigated pastures (also known as "lameiros"), small terraces used for the production of a wide variety of crops (e.g., potatoes, cereals), and the common lands, or "baldios" (a mixture of herbaceous species and shrubs often used for extensive grazing). Additionally, the presence of HNVf2 is reflected through com-

plexes of agricultural land mosaics of arable and horticultural crops mixed with vineyards, orchards and small woods, and even permanent pastures for grazing cattle [45,49]. The EDM has another characteristic production system, namely "masseiras", located on the coast, especially between Póvoa de Varzim and Esposende. These correspond to fields excavated in the dunes, in which the proximity of the large population centers of the AMP enables intensive horticulture management [53].

Concerning biodiversity conservation, the EDM has around 20% of its area enclosed in the EU Natura 2000 Network, encompassing 12 Sites of Community Importance (SCI) and 2 Special Protection Areas (SPA), the Transfrontier Reserve of Gerês-Xurés Biosphere and part of the Peneda-Gerês National Park. There are also 4 Important Bird Areas (IBAs), 7 National Protected Areas, and 1 wetland belonging to the "Ramsar Convention" [57].

### 2.2. Methodological Approach

To accomplish our goal, we developed a methodological approach in two stages (Figure 2). First, an ecologically based landscape and foodshed planning was applied to propose, for the first time, a potential foodshed relocalization scenario for the EDM. We start by mapping the land ecological suitability (steps 1 and 2; Figure 2) and selecting the ecologically suitable areas for agrarian land use. To accomplish this task, we employ the conditions established in the landscape potential planning for agrarian uses (Step 3; Figure 2), the ecological adequacy assessment regarding the current land occupation in 2018 (Step 4; Figure 2), and included an additional land use class for water and soil conservation (i.e., conservation forestry) (Step 5; Figure 2).

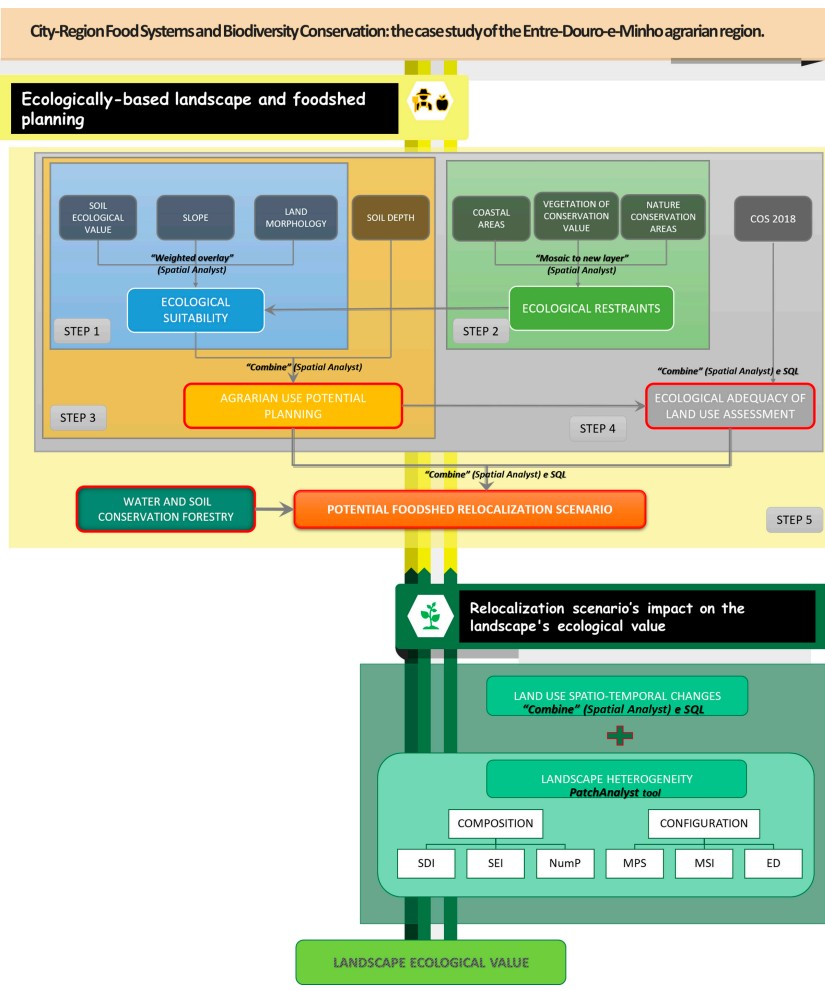

**Figure 2.** Overview of the methodological approach for the potential foodshed relocalization scenario and its impact on landscape ecological value.

Afterwards, in the second step, we evaluated the relocalization scenario's impact on the ecological value of the landscape in the EDM and the HNVf through the assessment of spatio-temporal changes in land use and landscape structure. This is the first study, to our knowledge, to complement CRFS and foodshed approaches with biodiversity impact analysis, such is the case of the computed landscape metrics proxies of heterogeneity and related Ecological Value assessment, also integrating the relocalization impact on HNVf, crucial for biodiversity conservation.

### 2.2.1. Ecologically Based Landscape and Foodshed Planning

The first step to propose the potential foodshed relocalization scenario demands a qualitative ecological suitability map (very high, high, moderate, reduced suitability) for agrarian usages (Step 1; Figure 2) based on an existing ecologically based landscape planning methodology [58–61] assuming landscape presents different ecological situations capable of supporting diverse human activities. The ecological suitability (ES) was determined according to the Multi-Criteria Decision Analysis approach (MCDA) developed by Cardoso et al. [17]. This approach allowed the assessment of three ecological criteria simultaneously, namely, the soil ecological value [62,63], the terrain morphology (tm) [64], and the slope [65,66] (Table S1a).

Afterwards, we readjust the resulting suitability map according to ecological constraints (Step 2; Figure 2) to safeguard critical components of the Ecological Network (i.e., green infrastructure; GI). According to Magalhães [61,67] the ecological network (EN) is recognized as a framework of ecological components providing the fundamental physical conditions for maintaining or restoring ecological functions and conserving and buffering core areas in terms of natural/semi-natural value while maintaining and establishing ecological connectivity. In this study, three EN components were included in the cartography as land use classes, namely: (1) coastal areas [68]; (2) nature conservation areas, matching the core areas for nature and biodiversity conservation, as the National System Classified Areas (NSCA; e.g., the "National Network of Protected Areas"—RNAP, Important Bird Areas—IBAs, "Natura 2000 Network—SCI and SPA", "UNESCO Biosphere Reserve", "Council of Europe Biogenetic Reserve", and the "Ramsar Convention" on International Importance Wetlands) [57]; and (3) vegetation with conservation value, corresponding to the natural and semi-natural vegetation from moderate to very high conservation value [69]. Concerning the readjusting process, in coastal areas land with moderate and higher ecological suitability was considered suitable for maintaining agrarian use.

The agrarian use potential planning (Step 3; Figure 2), representing the ecological conditions suitable for specific agrarian uses, namely temporary crops, permanent crops, pastures, and multiple agrosilvopastoral uses (Table 1), was developed according to the ecological suitability of the territory. The agrarian use assignment was carried out based on a reference bibliographic research concerning the characteristics of the specific usages in question [53,60,65,70]. For this task, a preliminary assessment was paramount regarding the distribution of agrarian land use classes according to the suitability base criteria (soil ecological value, slope, morphology), as well as the soil depth [71]. This step allowed the description of the agrarian land use features of the study area. It also enabled the adequate representativeness of the criteria defined in the potential landscape planning by taking into account the constraints determining land use in the EDM. Subsequently, based on the ecological conditions for each agrarian use class, we developed the synthesis of landscape potential planning for general agrarian use (Table 2). We must clarify that, in this work, the definition of multiple agrosilvopastoral uses corresponds both to heterogeneous agricultural areas and agroforestry systems (SAFs) (Table S1b).

**Table 1.** Landscape potential planning for the different agrarian usages (temporary crops, permanent crops, pastures, and multiple agrosilvopastoral uses) based on the ecological suitability of the EDM (2. Low suitability; 3. Moderate suitability; 4. High suitability; 5. Very high suitability; Soil ecological value 2. Low; 3. Variable; 4. High; 5. Very high).

| | Suitability | Soil Ecological Value | Slope (%) | Soil Depth (cm) | Notes |
|---|---|---|---|---|---|
| Temporary crops | 3 to 5 | 3 to 5 | <25 | >25 | |
| Permanent crops | 3 to 5 | 3 to 5 | <45 | >50 | Constraints according to the different slope classes * |
| Pastures | 2 to 3 | 2<br>3 | <16<br>16 to 45 | >25 | |
| Multiple agrosilvopastoral uses | 2 to 4 | 2<br>3 to 5 | <16<br>16 to 45 | >25 | |

* Constraints on agrarian practices: (a) slope 8% to 16%—crops according to contour lines; interleaved protection bands; level ditches for water infiltration, direct sowing, etc.; (b) slope 16% to 25%—construction of terraces; (c) slope 25% to 45%—possible agrarian practice when associated with soils with higher ecological value (>2) taking into account the characteristics of land use in the EDM region.

**Table 2.** Landscape potential planning for general agrarian use based on ecological suitability of the EDM (2. Low suitability; 3. Moderate suitability; 4. High suitability; 5. Very high suitability; Soil ecological value 2. Low; 3. Variable; 4. High; 5. Very high).

| | Suitability | Soil Ecological Value | Slope (%) | Soil Depth (cm) | Notes |
|---|---|---|---|---|---|
| General agrarian use | 2 to 5 | 2<br>3 to 5 | <16<br>16 to 45 | >25 | Constraints according to the different slope classes * |

* Constraints on agrarian practice: (a) slope 8% to 16%—crops according to contour lines; interleaved protection bands; level ditches for water infiltration, direct sowing, etc.; (b) slope 16% to 25%—construction of terraces; (c) slope 25% to 45%—possible agrarian practice when associated with soils with higher ecological value (>2) taking into account the characteristics of land use in the EDM region.

The ecological adequacy of land use (Step 4; Figure 2) comprised an intermediate procedure for the relocalization scenario allowing the adequacy assessment for the current land use by 2018 [17,67]. To accomplish this task, we compared the conditions established in the landscape potential planning for agrarian use with the land cover for 2018 (COS2018). This allowed the identification of areas where current agrarian use is according to land suitability or whether it should be changing in the potential relocalization scenario.

In the final step (Step 5; Figure 2) complementary scenarios for each specific agrarian use (i.e., temporary crops, permanent crops, pastures, and multiple agrosilvopastoral uses) were developed. Overall, following the landscape potential planning for agrarian uses we identify ecologically suitable areas for each of the referred uses. As a result, we identified areas with existing agrarian use to be maintained and additional areas, currently under different usages, to propose for agrarian use. Following the identification of agrarian ecologically suitable areas, land use patches classified as autochthonous forests (e.g., cork oak forests, holm oak forests, forests of other oaks, chestnut forests, and forests of other hardwoods) were held unchanged. Additionally, we proposed a water and soil conservation forestry class to improve the relocalization scenario, safeguarding forestry resources in the territory and associated ecosystem services. This class was considered to obtain the maximum potential for agrarian use only where ecological suitability and constraints allow it. The guidance criteria for its assignment are described in Table S1c.

Ultimately, the detailed foodshed relocalization scenario resulted from the specific agrarian usages assignment (e.g., temporary crops, permanent crops, pastures, multiple agrosilvopastoral uses). The scarcity of ecologically suitable areas for permanent crops in the EDM was notorious. For this reason, after allocating all the suitable existing crops (100%), the assignment criteria gave priority to all proposed permanent crops and only then proceeded to the allocation of proposed temporary crops and finally 50% of the pastures and 50% of the proposed multiple agrosilvopastoral uses.

For the presented methodology, we employed tools and spatial approaches in Geographic Information Systems (ArcGIS 10.7) [72]. We operated data from spatial databases accessed from the Epic-WebGis platform and the DGT (Directorate-General for Territory),

namely, the 2019 version of the Official Administrative Map of Portugal (CAOP) and the thematic map of land use and land cover for mainland Portugal for 2018 (COS2018). The cartographic datasets were obtained in the shapefile format, later reclassified, and converted into raster format, with a pixel of 25 m. The complete list of databases used is available for consultation in the Supplementary Material (Table S1d).

2.2.2. Relocalization Scenario's Impact on the Landscape's Ecological Value

The landscape's ecological value of the potential foodshed relocalization scenario was assessed by exploring land use dynamics and landscape indicators proxies of heterogeneity and biodiversity. Some authors have used "Ecological Value" and "Conservation Value" indifferently in the literature [30]. Regardless, while conservation value aims to quantify the ecosystem's potential to host rare or declining species prioritizing conservation efforts [73], in the present work, as aforementioned, the ecological value reflects the intrinsic value of the landscape, specifically related to the benefits that nature offers to sustain biodiversity [28].

The assessment focused primarily on agrarian land use classes (i.e., temporary crops, permanent crops, pastures, and multiple agrosilvopastoral uses) but also forests, shrubs, and vegetation (i.e., designated as "forest and other usages"; see Table S2a). This methodological step was carried out for the entire study area of the EDM agrarian region and the HNVf1 and HNVf2 [45]. The complementary analysis of HNVf is related to the crucial role of low-intensity agriculture in maintaining and supporting these landscapes. For this reason, the current approach is crucial to understand the impact of the potential relocalization scenario on these distinct landscapes vital for biodiversity conservation. The HNVf1 and HNVf2 cartography results from spatially explicit identification work in the EDM region developed by Lomba et al. [45] from several indicators integration, namely landscape structure and composition, the intensity of agricultural management, and diversity of crops. This work was supported by information available for 2009, the most up-to-date information, as recent temporal resolution data were not available during the development of this work (e.g., General Agricultural Census 2019).

Firstly, the assessment of the spatio-temporal changes in land use sought to explore the land use dynamics resulting from the relocalization scenario performance comparatively with the COS2018. The analysis was conducted on raster cartography (25 m pixel) using the Combine tool (Spatial Analyst) and Structured Query Language (SQL). This approach allowed us to identify the land use transitions between the two timelines Afterwards, we compute the land-use conversion area (ha) from the COS2018 to the scenario and the respective percentage.

Finally, to accomplish the spatial characterization of the landscape, we carried out a literature review supporting the identification and selection of metrics frequently used as proxies of landscape heterogeneity. According to Fahrig et al. [31], there are two components of landscape heterogeneity: compositional heterogeneity (the number and proportions of different cover types) and configurational heterogeneity (the spatial arrangement of cover types). The selected metrics were subsequently verified through correlation analysis, depicting results below 0.7 [29,45,74–77]. Three metrics related to composition (SDI, SEI, and NumP) and another three connected to configuration (MPS, MSI, and ED) were chosen (Table S2c). The metrics were computed in vector format maps with PatchAnalyst extension for ArcGIS 10.7 [72], at landscape and class level, for both COS2018 cartography and the relocalization scenario, at the scale of the EDM region and the HNVf (1 and 2).

## 3. Results

*3.1. Potential Foodshed Relocalization Scenario*

The ecological suitability (ES) assessment following the three ecological criteria (e.g., soil ecological value, terrain morphology, and slope) resulted in a spatially explicit map (Figure 3). The outcome represents the distribution of different suitability levels (e.g., very high, high, moderate, low, and very low), revealing the prevalence of land with moderate suitability (338,238 ha; 38.6%; i.e., essentially located on slopes above 8%), followed by

reduced (262,462 ha; 30%) and high (215,546 ha; 24.6%) ES. Only 1.6% (13,769 ha) of the territory shows very high ES.

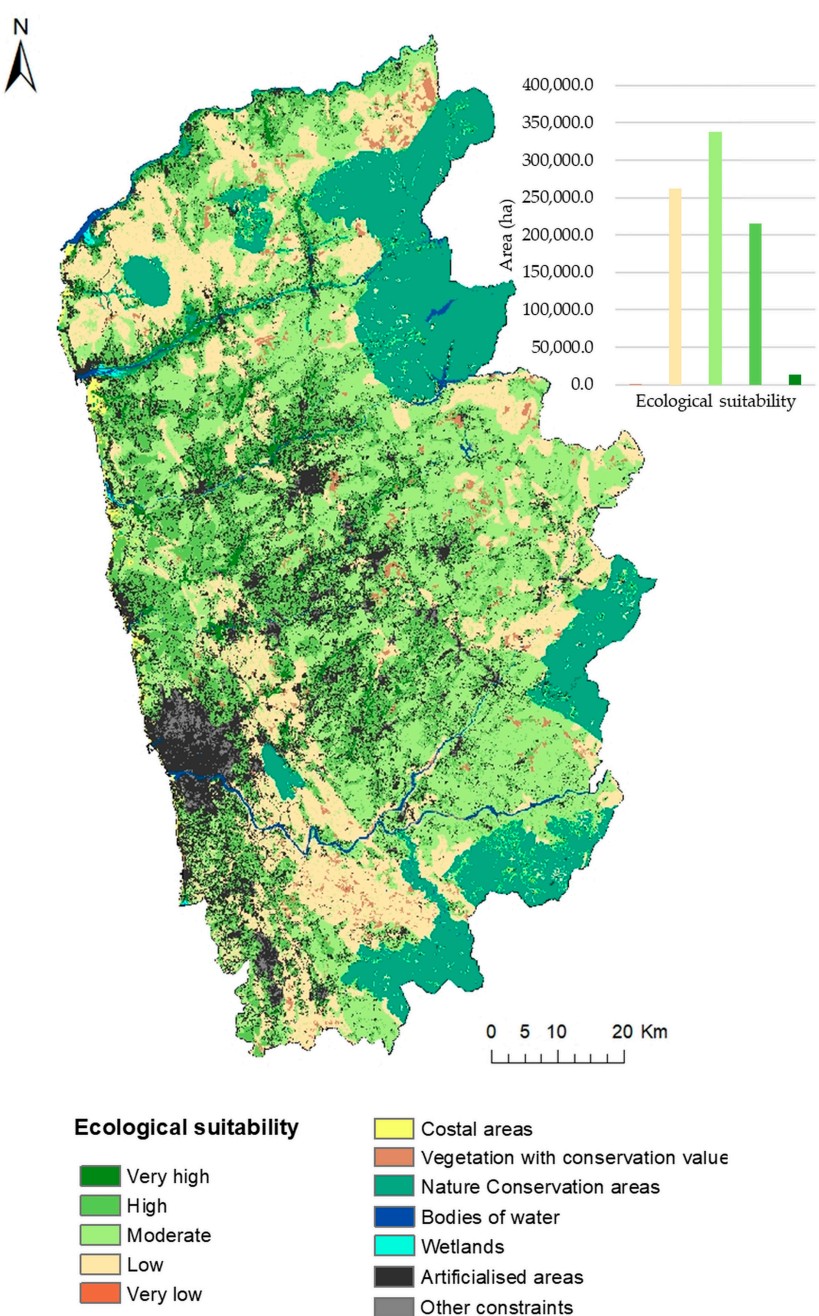

**Figure 3.** Ecological suitability in the EDM.

Additional analysis depicted a high percentage of artificialized areas (43%; 54,829 ha) overlapping land with high ES. Regardless, forests, shrubs, and vegetation classes, identified as the predominant land use in the EDM (539,282 ha), were found mainly in land with low (235,840 ha) and moderate (233,758 ha) ES. Agrarian areas, mostly composed of temporary crops, are predominantly located in areas with high suitability.

The ES analysis regarding the presence of ecological constraints (e.g., coastal areas, nature conservation, and vegetation with conservation value) revealed over 50% of areas under constraints match land with reduced ES, not competing with agrarian usages. However, 11.3% (20,660.3 ha) of agrarian usages in land with high and very high suit-

ability, mostly temporary crops (13,901.9 ha; 7.6%), are overlapping areas with ecological constraints, particularly in coastal areas.

The outcome of the ecological adequacy assessment for the existing agrarian use by 2018 revealed some discrepancies between general use and specific use. Particularly in the case of temporary crops (Figure 4), areas classified as inappropriate use (i.e., highlighted in red) were observed predominantly close to the coast in the AMP and northern areas. Several of these locations are characterized by soils with reduced ecological value (13,216.1 ha; 59%), thus not ecologically suitable for these crops, according to the landscape potential planning. Regardless, some other spots classified as inadequate in peri-urban location (2725 ha; 12.2%) resulted from limitations imposed by the cartography, namely the "social" designated areas in the soil mapping. Other specific usages, such as permanent crops, pastures and multiple agrosilvopastoral uses, are less represented in the EDM, exhibiting a dispersed and discrete presence, without adequate resolution on the map concerning inadequate usages.

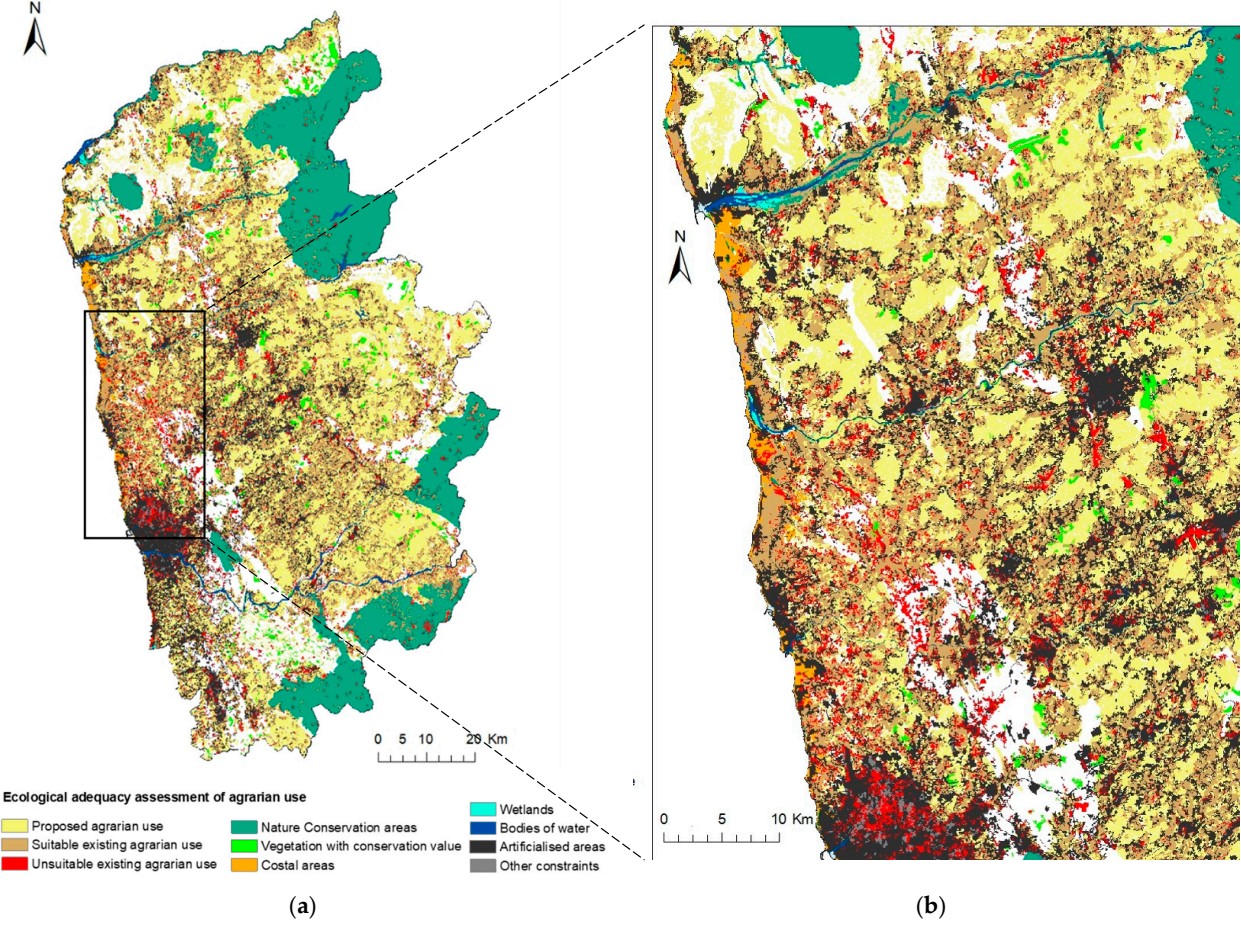

**Ecological adequacy assessment of agrarian use**

- Proposed agrarian use
- Suitable existing agrarian use
- Unsuitable existing agrarian use
- Nature Conservation areas
- Vegetation with conservation value
- Costal areas
- Wetlands
- Bodies of water
- Artificialised areas
- Other constraints

(**a**)     (**b**)

**Figure 4.** *Cont.*

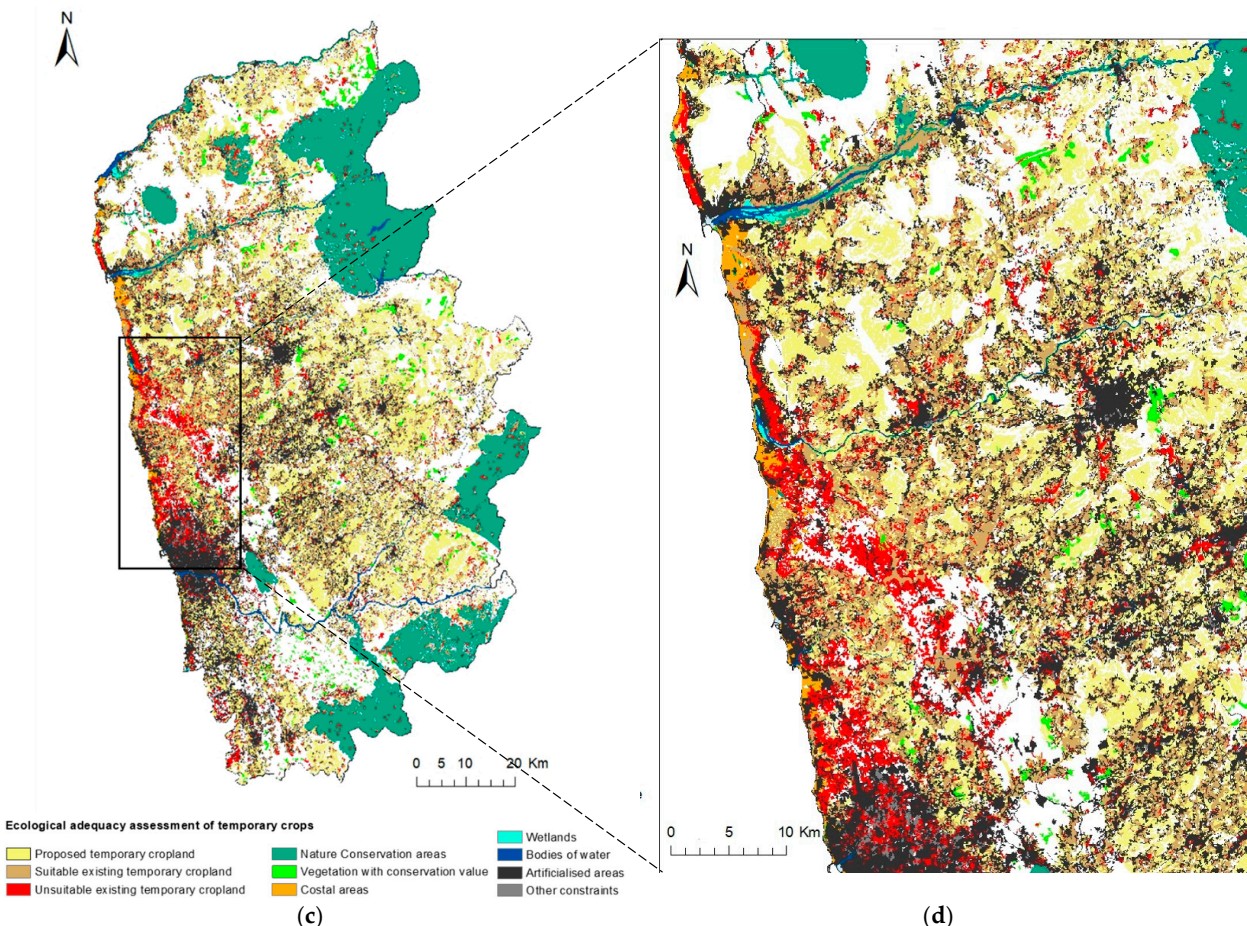

**Figure 4.** Ecological adequacy assessment (adequate or inadequate) of agrarian use concerning existing land use in 2018 (COS2018). The distinction between (**a**) general agrarian use and (**c**) specific agrarian use (temporary crops) with a detailed view located on the coast of the Porto Metropolitan Area and northern areas for both (**b**) general agrarian use and (**d**) temporary crops. The highlighted areas (in red) correspond to areas where existent agrarian use is classified as inadequate.

Finally, Figure 5 shows the result of the EDM potential foodshed relocalization scenario for the different agrarian usages, assigned accordingly to the ecological suitability criteria. In the relocalization scenario, existing temporary crops still represent the dominant land use with 21.3% (186,982.8 ha) of the EDM (Figure 6). These crops have the highest percentage of the proposed area corresponding to 9.9% (86,793.4 ha) of the EDM. Regardless of the notorious scarcity of ecologically suitable areas for permanent crops in the agrarian region, they undergo a significant increase of 7.3% (64,056.4 ha) of the proposed area, tripling their occupation to a total of 9.4% (82,400.3 ha), although distributed in a dispersed way, standing still very distant from the temporary crops. This might be explained by the permanent crops' requirements in terms of soil depth (minimum of 50 cm) [65], limited by the biophysical characteristics of the study area; i.e., a total of 63% has soil depth values lower than 50 cm. Additionally, we should remark that the increased percentage of temporary and permanent crops matches the maximum values according to the agrarian use potential planning and the suitability methodology developed. Regarding multiple agrosilvopastoral uses, it is also possible to mark a significant expansion to 8.7% (76,068.6 ha) of the EDM, corresponding to the addition of 5.8% (50,530.7 ha) of the proposed area. Finally, pastures continue to be the least represented, despite observing a significant increase, from just 0.1% (1152.4 ha) of the adequate existent area to 1.6% (14,307.8 ha) of the whole territory.

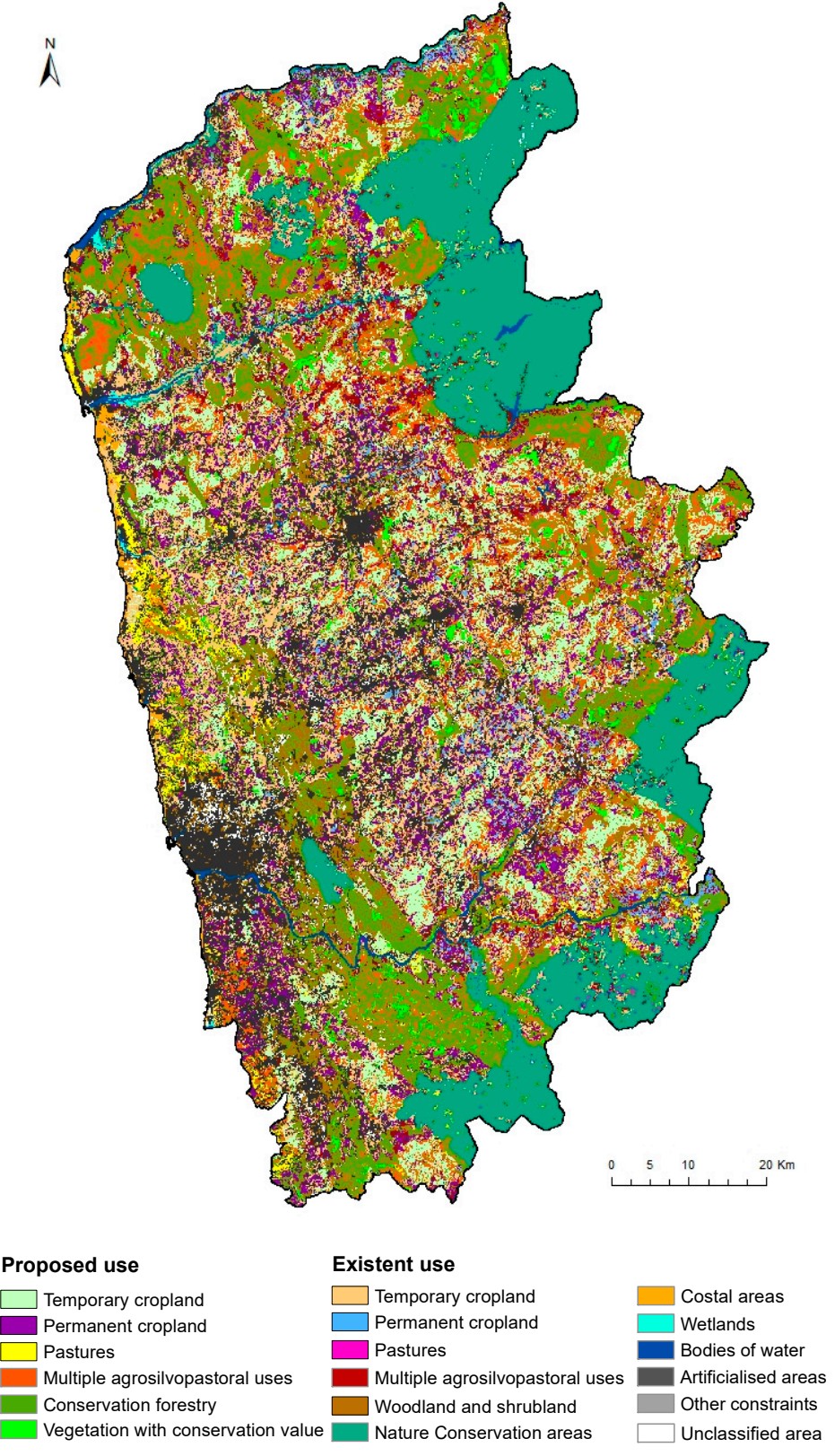

**Proposed use**

- Temporary cropland
- Permanent cropland
- Pastures
- Multiple agrosilvopastoral uses
- Conservation forestry
- Vegetation with conservation value

**Existent use**

- Temporary cropland
- Permanent cropland
- Pastures
- Multiple agrosilvopastoral uses
- Woodland and shrubland
- Nature Conservation areas
- Costal areas
- Wetlands
- Bodies of water
- Artificialised areas
- Other constraints
- Unclassified area

**Figure 5.** Potential foodshed relocalization scenario for specific agrarian usages in the EDM agrarian region (distinction between existing land uses that are maintained and new proposed usages). (1:550,000).

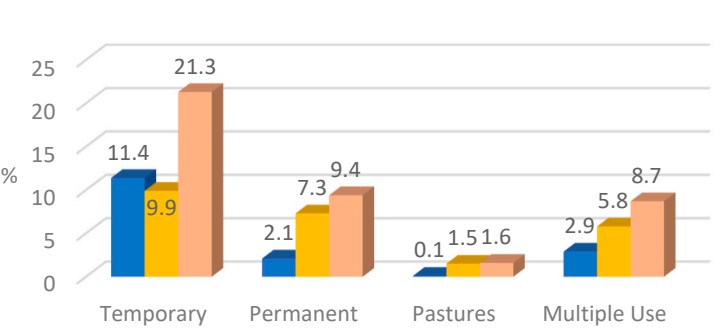

**Figure 6.** Synthesis of the land cover results regarding the different agrarian usages (temporary crops, permanent crops, pastures, and multiple agrosilvopastoral uses) in the EDM relocalization scenario according to the proposed, total, and adequate surface.

### 3.2. Spatio-Temporal Changes in Occupation and Land Use

In the EDM relocalization scenario, the outcome revealed a general increase in agrarian use by 18.3% (+160,834.3 ha) when compared to the COS2018, as would be expected (Figure 7). This is essentially the result of the conversion of *Eucalyptus* and *Pinus pinaster* forests and scrubland with agricultural potential in the COS2018 (i.e., 26.1%, 14.6%, and 11.8%, respectively; Figure S2a). Moreover, HNVf1 and HNVf2 assessments depicted contrasting dynamics. In HNVf1 there is an increase of 28.3% (25,524.9 ha) in agrarian use, essentially of temporary crops (Figure 8), which mainly results from the conversion of scrubland areas with agrarian suitability (i.e., 59.5% of the proposed agrarian area; Figure S2a). Conversely, in HNVf2, the trend is reversed, with an overall decrease of 8.3% (1717.6 ha) in agrarian use, mainly represented by temporary crops. This followed, in turn, by an increase in "forest and other usages" (i.e., mainly the proposed class of conservation forestry; Figure S2b).

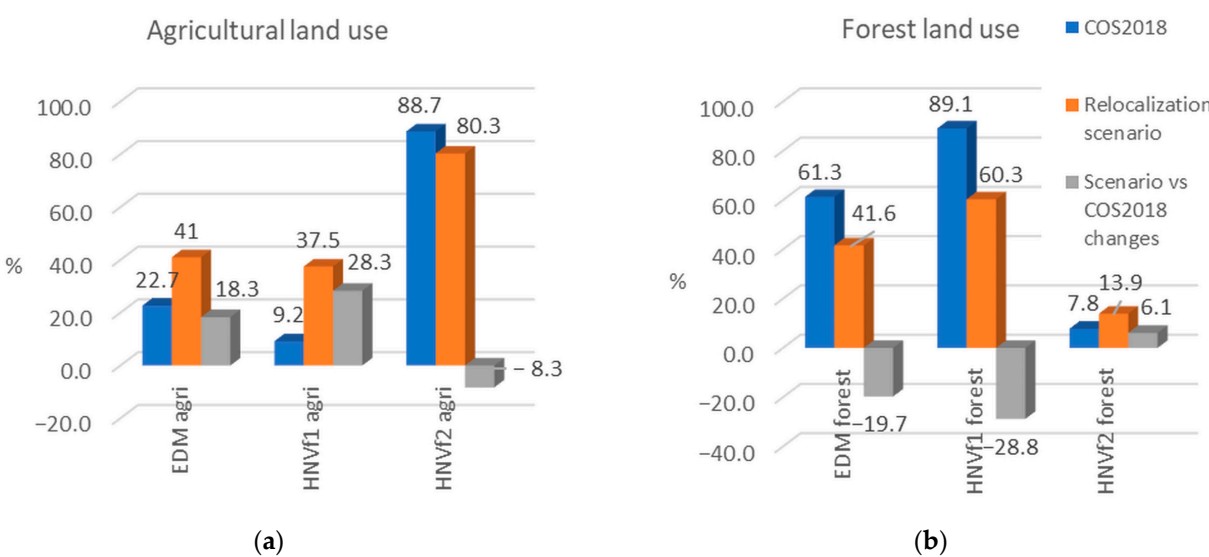

(**a**)                                                                              (**b**)

**Figure 7.** Quantitative analysis of land cover changes proposed by the relocalization scenario in the EDM and in the HNVf1 and HNVf2, concerning the 2018 cartography (COS2018) in (**a**) agricultural land use and (**b**) forest land use.

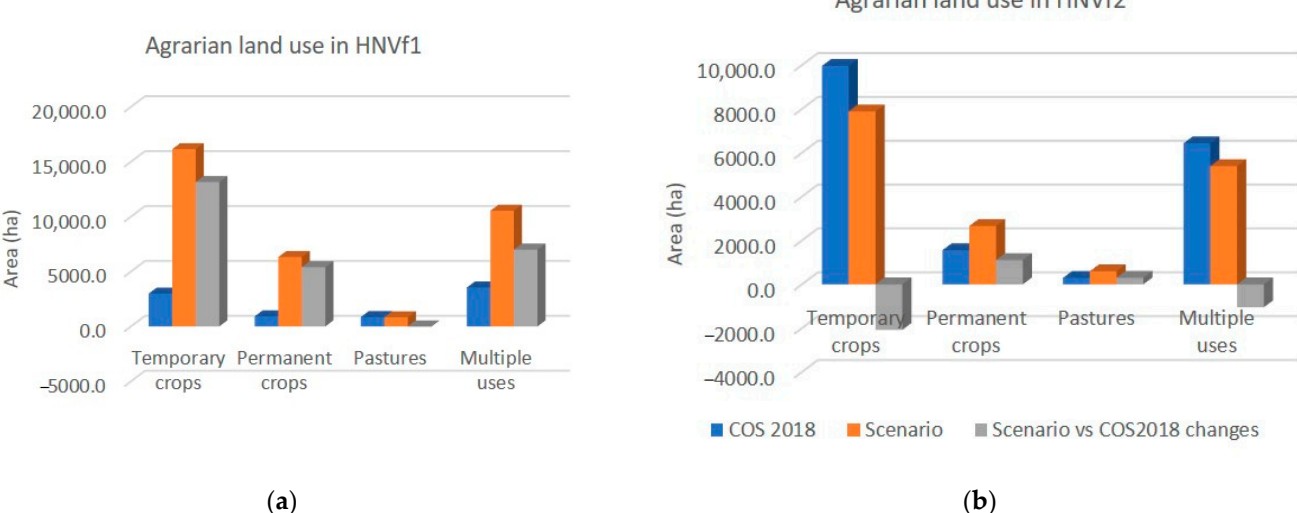

**Figure 8.** Comparative distribution of the different agrarian land usages presented in the COS2018 and the relocalization scenario, as well as the respective area of increase or decrease in both: (**a**) HNVf1 and (**b**) HNVf2.

Despite the agrarian increase in HNVf1, we must emphasize these areas continue to be highly represented by forest and other usages (60.3%, 54,330.9 ha), mainly composed of nature conservation areas (35.4%, 31,869.6 ha of the total HNVf1; Figure S2b). Additionally, in HNVf2, agricultural use is also maintained as the dominant land use (80.3%; 16,533.4 ha). Thus, the differentiating feature in land use between HNVf1 and HNVf2 remains in the relocalization scenario.

*3.3. Spatial Characterization of the Landscape*

Results concerning the landscape structure of the relocalization scenario mirror an increased landscape heterogeneity linked to an improved ecological value. Findings of the computed landscape metrics revealed a transversal trend, both in the EDM and the HNVf1 and HNVf2 (Table S2d), regarding an increase in diversity (SDI) and evenness (SEI) landscape indicators, reflecting characteristics of a more diverse landscape with a uniform distribution of land use classes. Additionally, we depicted a gain in the number of patches (NumP), accompanied by a decline in their average size (MPS) and an increase in edge density (ED) at the landscape level. It is worth noting the metrics results range is significantly higher in HNVf1 than in HNVf2.

At the class level, the results of the scenario's landscape structure analysis for agrarian land usage revealed the same tendency of increased patch number, reduction in the average size, and higher edge density (Figure 9). We noticed the apparent trend reversal of the declining patch size in HNVf1 concerning temporary and permanent crops. Regardless, we considered it negligible since patches continue to be small-sized (i.e., 1.6 ha and 0.8 ha, respectively). Although temporary crops equal the dominant agriculture land use in the EDM and HNVf1 and HNVf2, it is possible to observe a relatively increased number of patches representing permanent crops, which emerge more dispersed in the landscape in smaller size patches when compared to temporary crops. Multiple agrosilvopastoral uses also revealed high patch numbers, close to permanent crops, with decreased average size, although in HNVf2 they correspond to the larger patches. Finally, pastures are the least represented in the landscape, with few patch numbers, small sizes, and the lowest edge density.

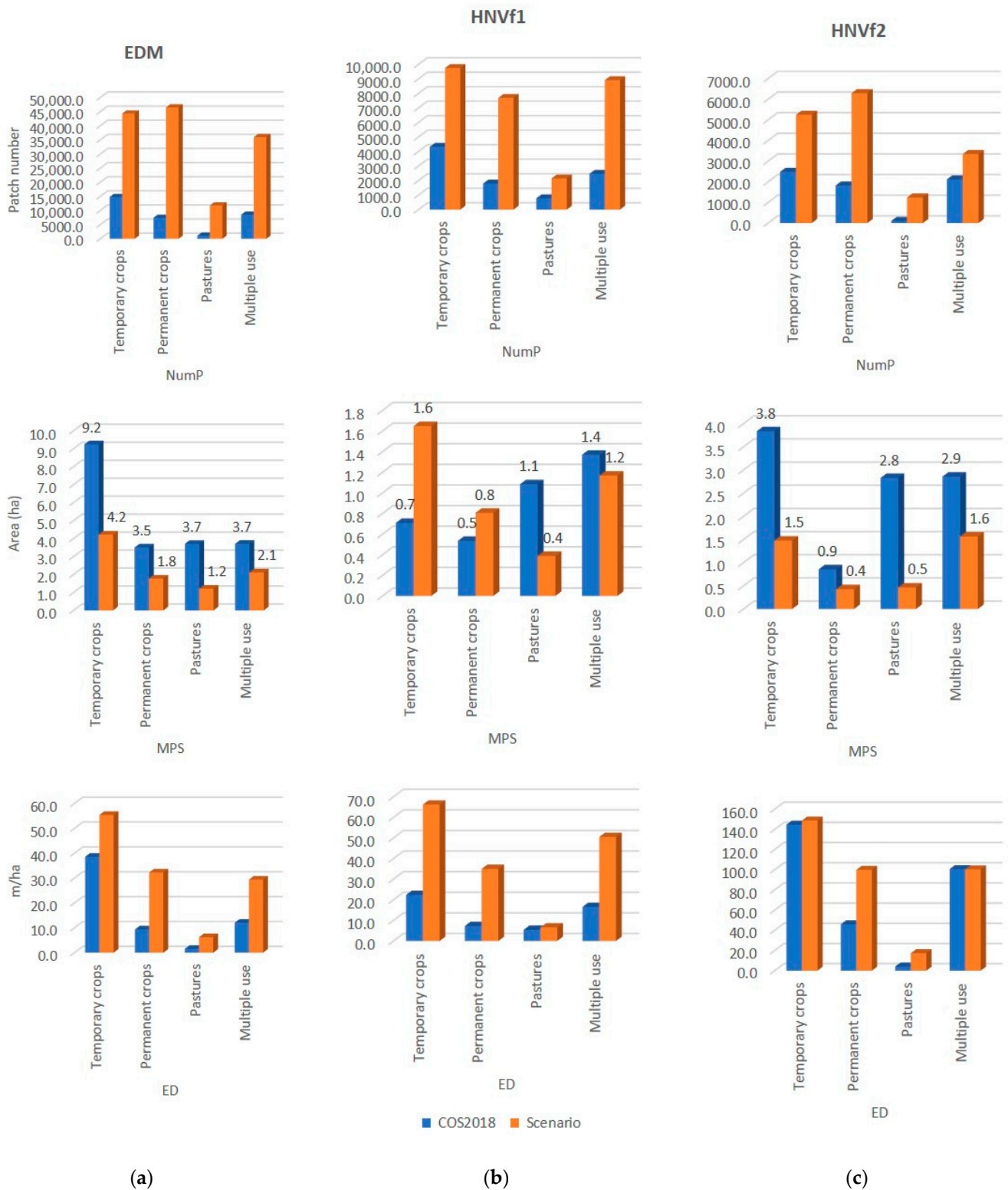

**Figure 9.** Comparative metrics results at the class level between COS2018 and the relocalization scenario for agrarian land use classes (temporary crops; permanent crops; pastures and multiple agrosilvopastoral uses) at the: (**a**) EDM and both (**b**) HNVf1 and (**c**) HNVf2 (CA: Class area, NumP: Number of patches, ED: Edge density, MPS: Mean patch size, ha: hectare, m: meter).

## 4. Discussion

We implemented an ecologically based landscape planning methodology to propose a potential foodshed relocalization scenario. Our aim was the ecological sustainability

of the landscape, where the proposed land uses consider ecological suitability ensuring their adequacy to the landscape's biophysical characteristics [78]. Results revealed only 1.6% of the territory presents very high ecological suitability, with a large part of the land with moderate suitability (i.e., essentially located on slopes above 8%). Therefore, the regional topographical characteristics imply constraints on agrarian land uses that require specific management practices (e.g., tilling along contour lines, buffer strips, level ditches for water infiltration, direct seeding, etc.) [60,65]. Additionally, a significant percentage of artificialized areas in high suitability land, together with the high population density of the territory, has been increasing the pressure on rural territories, namely those with increased agricultural suitability, which could be responsible for substantial agriculture losses [54]. This process of urban expansion associated with the ancestral tendency of settlements in areas of increased agricultural suitability has been affecting European countries, playing a considerable impact on the agrarian sector, consequently risking food security [79,80]. In the last 30 years, there have been significant changes in Portugal's land use; the 62.83% increase in artificial territories and the 12.48% decrease in agricultural areas are noteworthy [54]. Considering that soils with a high capacity for biomass production are scarce in Portugal and perform both economic and ecological functions (regulatory, support, and provisioning ecosystem services), it is crucial to reflect on the importance of preserving soils with higher ecological value and more suitable for agricultural activity.

The landscape potential planning for agrarian uses was built under the principle of land resources optimization, reflecting the concept of the best ecological suitability conditions for agrarian use and other landscape planning criteria. Moreover, results highlighted discrepancies between the ecological adequacy (i.e., assessment for the current land use by 2018) of general agricultural use vs. specific uses, further reinforcing the importance of ecologically based planning according to specific agricultural land uses (e.g., temporary and permanent crops). Additionally, the ecological adequacy assessment of agrarian use allowed pinpointing and replacement of particularly inappropriate occupations. This is the case of temporary crops located in soils with reduced ecological value with probably considerable impacts on the ecosystem conditions in coastal areas. This way we aimed at safeguarding natural resources and the most ecologically sensitive areas. The coastal areas are a first-level component in the Ecological Network (EN) in Portugal, due to their characteristics of high ecological value, sensitivity, and biological productivity and where the EN mapping process identified the overlapping of five or seven first-level EN components [81]. In Portugal, about 75% of the population lives near the coast, and this has been exerting significant pressure on coastal areas. Competition for land triggers the search for efficiency [82], and agricultural land management can compromise coastal ecosystem functions, in the absence of appropriate regulations, monitoring, and evaluation of land use. Moreover, *masseiras* are an interdune agroecosystem existent along the northern coast of Portugal, between Esposende and Vila do Conde, where the advantage of the shallow coastal aquifers, terrain modeling for wind protection, with vineyards plantation, and the addition of organic matter, such as seaweed and crabs' residues, allowed increasing production despite poor sandy soils and harsh environmental conditions [83]. These traditional agroecosystems, considered for their sustainable polyculture, have however been subjected since the 1960s to agricultural intensification. Conversion of these areas to "enlarged masseiras" with greenhouse vegetable production and use of chemical fertilization and pesticides have contributed to aquifer depletion, stressing the need to change the production system [84]. In the scope of Coastal Zone Management Programs (POOC), a type of water resources and land use plan, and Municipal Master Plans (PDM), that is Land-use Plans at the local scale, it may be necessary to assess what policy instruments and land use measures are necessary to steer improved land use and management. This is of particular importance so that agricultural land uses that entail environmental degradation or that increase vulnerability to coastal hazards are regulated and restricted. To promote their sustainable land use, these areas must be further evaluated on a larger scale and with additional methodologies that

consider, for example, farming systems and agricultural management practices and their degree of intensity or potential for environmental degradation.

Concerning the scenario's impact, previous research [27,85] has stressed the need to link the farming systems perspective with landscape ecology patterns and processes (i.e., land-use patterns and landscape ecological functioning) to maintain high levels of biodiversity when designing agroecological landscapes. Likewise, Tello and González de Molina also emphasized the role of biocultural heritage on resilient agroecosystems, such as the case of HNVf. In this way, we aimed to address such a gap by presenting data that indicate CRFS can operate as a fundamental concept to improve landscape planning and land use dynamics, fostering landscape heterogeneity and ecological value consistent with biodiversity conservation. Our results portrayed a potential foodshed relocalization scenario associated with an eminently agrarian landscape with improved ecological value.

Findings regarding land use changes revealed an increased agrarian use throughout the EDM, with a remarkable presence of temporary crops in the landscape and significant growth in permanent crops. Analyzing transition dynamics, we found that a large part of these agricultural areas resulted from the conversion of eucalyptus forests located in soils with agrarian suitability. It is critical to note that eucalyptus (i.e., *Eucalyptus globulus)*, currently occupying a significant area in the Portuguese forests [57] and 20.5% of the study area, is an exotic species with several ecosystem disadvantages, namely, high water consumption and erosion capacity of soils, with little potential in terms of ecological value and biodiversity [86,87]. Moreover, in terms of forest types, conifer and eucalyptus plantations have more susceptibility to fire compared to mature forests of broadleaved deciduous vegetation and mixed forests [88]. Therefore, the scenario proposal for its conversion to agricultural use in suitable land comprises a favorable aspect of the landscape's ecological value. However, its effect on biodiversity will depend on the created landscape mosaic, the production systems, and respective agricultural practices. Additionally, the EDM dominant land use in the relocalization scenario still corresponds to the group of forests and other usages, mainly integrated into the proposed land use class—water and soil conservation forestry. This proposed land use should be an autochthonous forest, with native and archaeophytes species, known for its ecosystems of high ecological value, that would play a crucial environmental and ecological role, both related to water and soil conservation and biodiversity protection [78,86,87].

The HNVfs assessment of land-use changes and dynamics depicted HNVf1 as still mainly occupied by nature conservation areas reflecting a higher percentage of natural and semi-natural vegetation (i.e., shrubs and sparse vegetation). While HNVf2, on the other hand, is essentially agrarian. This marked difference in land use occupation between HNVf1 and HNVf2 comprises a differentiating feature [45,47,48] that remains in the relocalization scenario. However, it is crucial to clarify that even if outside the limits of the designated areas for nature conservation, the HNVf1 proposed agrarian use should always be conditioned by the criteria underlying the practice of extensive agriculture characteristic of HNVf (i.e., reduced inputs of nutrients and fertilizers, in small plots, associated with the creation of landscape mosaics) [45,47,48]. Moreover, the proposed agrarian areas can also have a relevant function to mitigate fire hazards, since research identified that agricultural land cover types (agricultural crops) are less fire-prone, mainly for their low combustibility in comparison to other land cover types [89]. Concerning HNVf2, the relocalization scenario proposes a decrease in agriculture, mainly at the expense of temporary crop reduction. This decrease is, in turn, associated with an increase in forest use, mostly consisting of conservation forestry. The inclusion of this class in HNVf2, ideally composed of autochthonous forest, could also be a positive aspect, diversifying the landscape mosaic by the inclusion of forest ecosystems, contributing to increasing the landscape heterogeneity and safeguarding its ecological value [28,90].

The new landscape structure proposed by the relocalization scenario foresees an increase in landscape heterogeneity, which may mitigate some of the adverse effects of more intensive land use at the local level [91]. Increasing configurational heterogeneity

(number, size, and arrangement of patches) will expand the variability of conditions and ease species movement across the landscape, as other researchers attest [31,34,42,92,93]. Additionally, the higher number of patches of a smaller size may feature a characteristic related to less intensive forms of management. Likewise, the compositional heterogeneity, reflecting the variety of land uses, whether agrarian or natural, is also expected to be positive for the scenario's ecological value and biodiversity as it will enable a variety of conditions for a higher number of species with contrasting ecologic requirements, thus supporting more species than landscapes with homogeneous matrices [28,40–42,94,95]. However, we must bear in mind that excessive habitat heterogeneity may also have adverse effects, increasing habitat fragmentation at the expense of specialist species, as pinpointed by previous studies [96,97].

It was also possible to verify an increased variation range in the landscape metrics for the HNVf1 compared to the HNVf2. This could be related to the fact that HNVf1 landscapes are dominated by natural and semi-natural vegetation, where the scenario can add significant diversity by increasing small patches of the proposed agrarian usages, thus resulting in a more heterogeneous landscape. This is particularly relevant if we account for the dynamics of agricultural land abandonment and afforestation, responsible for the gradual decline in the diversity and complexity of the landscape, which has been affecting the traditional rural landscapes [37,45]. In the case of HNVf2, the range in the computed metrics is significantly smaller, possibly because they correspond to landscapes where the mosaic is already quite heterogeneous. Nevertheless, HNVf2 may also benefit considerably from the relocalization scenario, where the introduction of forest land use patches can add ecological value, improving the connectivity of the landscape mosaic [40,98].

Finally, it is important to reflect on some issues to be improved and possible lines of inquiry to follow in future work. The next step is to use the relocalization scenario to determine the self-sufficiency degree of the potential foodshed, considering several diets. Other allocation models can be employed and developed through the inclusion of diets that allow for the assessment of plant and animal production needs to obtain a degree of self-sufficiency according to the food needs of the resident population [16,17]. Also, alternative suitability methodologies can be explored to assess ecological suitability in the absence of soil mapping data (i.e., "social area"), particularly in urban and peri-urban agriculture areas. Furthermore, other assessment methodologies complementary to the land use adequacy could also be considered, especially for specific locations such as the coast, where the presence of land with high suitability is to decide on the conversion to agricultural use, due to the high sensitivity and ecological productivity of these areas. Also, including participative approaches, namely workshops and questionnaires, could help reveal people's framing of the regional food system and forecast their acceptance of relocalization.

Regarding the scenario environmental impact, for this work, we only intend to employ the landscape structure analysis as proxies of landscape heterogeneity crucial for the maintenance of the ecological value of the landscape and the associated biodiversity niches [41,93,99]. However, to establish a more direct relationship with biodiversity, additional studies could include species richness data. The influence of cartography classification is another element to consider [100], so it will be necessary to review and improve the categories used, particularly forest usage, to obtain more accurate results.

Moreover, the GI outlined at the national level [60] includes natural and semi-natural vegetation with conservation value and nature conservation areas, which also integrate agroecosystems of interest for biodiversity conservation (i.e., overlap with HNVf). New agricultural areas with favorable landscape metrics can guide the extension of GI by establishing this objective/criterion as a guide for foodshed relocalization scenarios. Eventually, it will be relevant to understand whether the apparent heterogeneity increase at the global landscape level could still be associated with more specialized and homogeneous landscapes in certain regions, which could be assessed by implementing a closer grid analysis (1 km × 1 km) for a more detailed study of the territory.

## 5. Conclusions

The research enabled the proposal of a potential foodshed relocalization scenario in line with ecologically based landscape planning criteria and ecological suitability for agrarian uses. Findings regarding land use changes disclosed an increase in the extension of agrarian land uses in the region, with significant growth in the area allocated to temporary crops and moderate growth in the area of permanent crops. Land use dynamics assessment reveals that a large part of these agricultural areas resulted from the conversion of eucalyptus forests located in land suitable for agricultural uses. This land use conversion comprises a favorable aspect of the relocalization scenario proposal. However, its effect on biodiversity will depend on the created landscape mosaic, the production systems, and respective agricultural management practices. Other unsuitable existent agrarian land uses are proposed for conservation forestry, with autochthonous species, to be included in landscape ecological restoration projects. Moreover, the new landscape structure proposed by the relocalization scenario foresees an increase in landscape heterogeneity, both in composition and configuration, allowing a variety of conditions for different species with contrasting ecological requirements. Overall, our findings revealed that the proposed scenario constitutes an agrarian landscape of increased ecological value. Therefore, we provide a methodological approach for the proposal and assessment of potential foodshed relocalization scenarios, which, in the future, may be considered in land use planning instruments and food policy and planning, at regional and local levels. These should aim to enhance landscape ecological value while creating conditions for greater regional food self-sufficiency.

**Supplementary Materials:** The following supporting information can be downloaded at: https://www.mdpi.com/article/10.3390/su15065021/s1, Supplementary material (S1)—Ecologically based landscape and foodshed planning and Supplementary material (S2)—Relocalization scenario's impact on the landscape's eco-logical value.

**Author Contributions:** Conceptualization, A.S.C.; methodology, A.S.C. and A.L.; software, M.F., A.S.C. and A.L.; formal analysis, M.F.; validation, M.F., A.S.C. and A.L.; investigation, M.F., A.S.C. and A.L.; data curation, M.F., A.S.C. and A.L.; writing—original draft preparation, M.F.; writing—review and editing, M.F., A.S.C., A.L. and J.P.H.; visualization, M.F.; supervision, A.S.C., A.L. and J.P.H.; project administration and funding acquisition, A.S.C. All authors have read and agreed to the published version of the manuscript.

**Funding:** A.S.C. was funded by Foundation for Science and Technology (FCT) through the employment contract CEECIND/02726/2018 CP1546, granted in the scope of the Individual Scientific Employment Tender. A.L. and J.P.H. contribution to this research was supported by project MAGIC—Multi-Agent Control and Estimation for Multi-Horizon Goals Conciliation (POCI-01-0145-FEDER-032485) funded by FEDER via COMPETE 2020—POCI and by FCT/MCTES via PIDDAC. A.L. is supported by national funds through FCT—Fundação para a Ciência e a Tecnologia, I.P., in the context of the Transitory Norm—DL57/2016/CP1440/CT0001.

**Institutional Review Board Statement:** Not applicable.

**Data Availability Statement:** Not applicable.

**Conflicts of Interest:** The authors declare no conflict of interest.

## Notes

[1]  NUTS is an acronym for "Nomenclature Units for Territorial Statistics" established by Eurostat to provide a single uniform breakdown of territorial units for the production of regional statistics for the European Union (EU). The nomenclature encloses 3 levels (i.e., NUTS I, NUTS II, and NUTS III) defined according to population, administrative and geographical criteria. NUTS III correspond to the territory of Intermunicipal Entities. In 2015 a new regional division took place in Portugal—NUTS 2013.

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
