# Peer review of "City-Region Food Systems and Biodiversity Conservation: The Case Study of the Entre-Douro-e-Minho Agrarian Region"

_sustainability, doi:10.3390/su15065021_

Round 1

Reviewer 1 Report

I have read with great pleasure the article titled "City-Region Food Systems and Biodiversity Conservation: the case study of the Entre-Douro-e-Minho agrarian region". The study presented in the manuscript is scientifically sound and mature, and the research deep enough. The manuscript is of potential interest to a broad international audience, and well written. It deserves to be published by "Sustainability". However, there are some minor comments, aimed at improving the perception of manuscript after its publication.

Methodology: it would help presenting the reasons for selecting the case study. Although some of its characteristics offer good hints, such as its population density (lines 197-198) and land use conflict between agriculture and urban land uses (lines 200-201), the region was probably used as a typical case of "something" that needs to be defined in a bold way, justifying its choice.

The Discussions do a great job in presenting the significance of findings, methodological limitations of research, and future research direction, but need some more input on comparing the findings of those from other studies carried out elsewhere and stressing out more the contribution of results to the advancement of the field, in line with what is excellently pointed out in the abstract: "first attempt to map the relocalization of the potential foodshed" (line 15). In addition, the novelty of the methodology should be also emphasized.

I also recommend authors to pay more attention to the inner consistency of their manuscript. For example, figures are referred to even in the same parts of the text as "figure" (line 406) of "Figure" (line 409) - different capitalization, and references do not seem to match the formatting of the journal, and show inner inconsistency - for example, the name of journals is sometimes spelled out (reference #13), in other cases abbreviated as required (#12); volume number is sometimes introduced by "vol" (#78), in other cases without, as required (#79) etc. Finally, I advocate removing the period (.) concluding the title.

Reviewer 2 Report

Comments: City-Region Food Systems and Biodiversity Conservation: the 2 case study of the Entre-Douro-e-Minho agrarian region

First of all I congratulate authors for this excellent work on CRFS with wider insights for policy and research. The thoughts are well organized throughout the manuscript with proper citation and support from previous work. In addition, authors have amply contextualized the work to the ensuing issues related to climate change, population growth and industrialization. The introduction encompasses sufficient background, focus, rational and objectives of the paper. However, it could trimmed a little by rearranging some redundant information such as that related relocalization. This sentence in the subsection 1.1 needs a revisit in terms of use of words and syntax 'The main research question we intend to address in the present study is whether there is possible??? to obtain standards towards improved food self-sufficiency, capable of reconciling agricultural production and biodiversity conservation at the regional scale.'

The methods provide a clear background of and selection procedure of the study area. the approach of the paper has been extensively worked out and presented with clarity. A two-stage methodological approach has been adopted to reach out to the conclusions. The method seems to be devoid of integrating social aspects as well as population growth and people's perceptions and/or their expectation or constraint to opt relocalization of food systems as there can be a range of issues related to infrastructure and communication along with input availability and labor/machinery. The results would have been much enriched if some socio-demographic aspects could highlight the ease or issues related to geographical relocation of food system. Nevertheless, this aspect can be a good starting point for future work. the rest of the findings are highly intuitive, insightful and valid as well as worthy of reproduction in other regions of the world.

Reviewer 3 Report

Agriculture is the dominant form of land management and at least half of the species in Europe depend on agricultural habitats. There, research on urban-regional food systems and biodiversity conservation is of great importance.

However, the data in this manuscript derived from existing data products, and there is a lack of measured data or survey data.

Moreover, the methodology used in this manuscript, which is very well established, lacks innovation.

Therefore, rejection of the manuscript is recommended.
